# Effect of Tannin Furanic Polymer in Comparison to Its Mimosa Tannin Extract on the Growth of Bacteria and White-Rot Fungi

**DOI:** 10.3390/polym15010175

**Published:** 2022-12-29

**Authors:** Jonas Eckardt, Gianluca Tondi, Genny Fanchin, Alexander Lach, Robert R. Junker

**Affiliations:** 1TESAF Department, University of Padua, Viale dell’Università 16, 35020 Legnaro, Italy; 2Forest Products Technology and Timber Constructions Department, Salzburg University of Applied Sciences, Marktstrasse 136a, 5431 Kuchl, Austria; 3Evolutionary Ecology of Plants, Biodiversity of Plants, Department of Biology, University of Marburg, Karl-von-Frisch-Str. 8, 35043 Marburg, Germany; 4Department of Biosciences, Paris-Lodron-University Salzburg, 5020 Salzburg, Austria

**Keywords:** flavonoid, degradation, bio-plastic, antimicrobial, biological resistance

## Abstract

Tannins are well-known to protect plants from bacteria and fungi, but nothing is known about its effects on microorganisms once they are copolymerized. Therefore, a study was conducted to evaluate the effect of a tannin–furanic polymer in comparison with industrial mimosa tannin extract on the in vitro growth of two strains of bacteria, *Bacillaceae* and *Pseudomanadaceae,* and two white-rot fungi, *Trametes versicolor* and *Agrocybe aegerita*. Results have highlighted that the tannin polymer did not inhibit the growth of tested bacteria and even favored the growth of *Bacillaceae* without extra glucose. The growth of both fungi was enhanced by mimosa tannin and its polymer at low concentrations (<1%), while concentrations above 10% had a growth-inhibiting effect, which was slightly less strong for the polymer compared to the tannin against *Trametes versicolor.* These findings highlighted that tannin–furanic polymers can be tolerated by certain microorganisms at low concentration and that their inhibitory effect is similar or slightly lower than that of the pristine tannin extract.

## 1. Introduction

Combined with physical protection mechanisms, superior plants also synthetize specific molecules to protect themselves against biotic and abiotic attack, known as extractives [1]. It is known from the literature, that within the extractives, polyphenols such as tannins are particularly effective against fungi, bacteria and insects [2,3,4], as well as against abiotic stress such as oxidation and UV-rays [5,6].

The possibility to polymerize bio-molecules to higher molecular mass and high water resistance promoted the tannin extract as a possible sustainable precursor for bio-plastics and, hence, highly attractive for research in material science. In particular, the tannin–furanic polymers were produced by combining the tannin extract with another bio-based resource, furfuryl alcohol, in a polycondensation reaction occurring in an acid environment [7]. The polymerization of furfuryl alcohol with condensed tannins was already proposed in previous works and it is schematized in Figure 1.

Tannin–furanic polymers have already been considered as wood adhesives or bioplastics [8,9,10] and are, furthermore, of particular interest for the production of rigid foams [11,12]. These lightweight porous materials showed good thermal insulation (between 24 and 44 mW m^−1^ K^−1^) and good fire resistance properties (self-extinguishes) [13,14,15], presenting an attractive scenario as an innovative, greener bio-based alternative for building insulation. This opportunity has encouraged research into this material, which was upscaled to a pilot plant [16]. Unfortunately, the industrial production has not yet been implemented, also because of the unclear end-of-life usage and biocompatibility of the tannin–furanic polymer.

Until now, some potential end-of-life cases for tannin–furanic foams were considered, including (i) recycling the exhausted powder in fresh tannin foam formulations, (ii) use of grinded powder as an ammonia mitigator in cattle slurry, (iii) reusing the grinded foam as a pollutant adsorber for heavy metals, cationic dyes and anionic surfactants in wastewater treatment and (iv) thermal valorization [16,17,18,19,20]; however, major concerns remain. In particular, the use as an ammonia mitigator could be a promising application of these materials, helping to achieve the European targets for a sharp reduction in environmentally harmful ammonia emissions by 2030, which are still far from being met [21]. However, to realize such potential application in the agricultural sector, the biocompatibility of such polymers needs to be investigated.

The interaction of condensed tannin with bacteria and fungi has already been investigated by several authors. In the review work of Ogawa and Yazaki, the antimicrobial activities of mimosa tannin extract against several bacteria such as *E. coli, K. pneumoniae, P. vulgaris* and *S. marcescenes,* and fungi such as *Coriolus versicolor*, *Tyromyces palustris* and *Aspergillus niger* [22] were summarized. The study tendentially converge in considering mimosa tannin extract as a good antimicrobial substance. These findings were also observed by Peng et al. where condensed tannin inhibited the growth of total bacteria, yeast and fungi [23]. However, the condensed tannins are molecules produced by nature and, despite being highly recalcitrant, there are also certain strains of bacteria and fungi that can metabolize them through specific enzymes (e.g., laccase, oxygenases, tyrosinase, tannase) [24], as more recently reported in the investigation of Prigione et al. with filamentous fungi [25].

In this study, the growth behavior of tannin-tolerant bacteria and fungi in the presence of tannin extract and tannin–furanic polymers was observed to understand whether tannin loses its antimicrobial properties once polymerized. In vitro growth of bacteria and fungi in presence of the two powders were monitored. The growth of two bacteria of the family *Bacillaceae* and *Pseudomanadaceae*, which were isolated from tannin–furanic foam cubes that were previously buried in a compost heap, was analyzed with and without glucose as additional carbon source. The same powders were also tested on the growth of two common white-rot decay fungi, (*Trametes versicolor* (L.) *Lloyd* and *Agrocybe aegerita* (V. Brig.) *Singer*).

## 2. Materials and Methods

### 2.1. Materials

Commercial mimosa (Acacia mearnsii, De Wild.) tannin (MT) powder from Tanac (Montenegro, RdS, Brazil) was used as the reference for this study. The general composition of the tannin extract is provided by the supplier, and detailed spectroscopic and chromatography analysis was published in former years [6,7,26]. Formaldehyde-free tannin–furanic foams were obtained by the copolymerization of mimosa tannin and furfuryl alcohol (62:38 wt%) under acid conditions, referred to as standard formulation in previous research [27]. Briefly, 126 g mimosa tannin (MT) was mixed with 77.7 g furfuryl alcohol, 25.4 g of water and 16.5 g diethylether until homogenization. In a second step, 54.4 g of sulfuric acid (32%) was added, and the mixture was poured into a Teflon mold of 30 × 30 × 4 cm^3^ before undergoing hot-press curing at 90 °C for 30 min. The cured foams were stabilized at 20 °C and 65% m.c. for at least one month before undergoing the biological tests against bacteria and fungi. The foams were directly used for the colonization/isolation of the bacteria. For the different growth tests, the subject of this study, the foams, were grinded and sieved into fine powder (TFP) of <1 mm granulometry. The powder was then washed with distilled water to remove the remaining acid catalyst. Both powders were sterilized by an autoclave before testing. In Figure 2, the visual appearance of the two powders is reported.

For the isolation of bacteria, a phosphate-buffered saline (PBS, Carl Roth GmbH + Co, Karlsruhe, Germany) and 800 mL of a standard medium (SM) of H_2_O containing 16 g LB broth (A. Hartenstein; Darmstadt, Germany), 12 g agar (Carl Roth GmbH + Co.; Karlsruhe, Germany) and 0.8 g glucose (E. Merck; Darmstadt, Germany) were used. After autoclaving 24 mg cycloheximide (Sigma-Aldrich Chemie GmbH, Taufkirchen, Germany) were added to the standard medium (SM) to prevent fungal growth. Isolated bacteria were of the families *Bacillaceae* and *Pseudomanadaceae*. For the bacterial growth essay, a concentrated salt solution was mixed with 30 g Na_2_HPO_4_ (E. Merck; Darmstadt, Germany), 15 g KH_2_PO_4_ (Carl Roth GmbH + Co.; Karlsruhe, Germany), 2.5 g NaCl (Carl Roth GmbH + Co.; Karlsruhe, Germany) and 5 g NH_4_Cl (E. Merck; Darmstadt, Germany) and filled up with 500 mL H_2_O. Next, 10 mL of the salt solution together with 10 μL of CaCl_2_ [1 mol∙L^−1^] (E. Merck; Darmstadt, Germany) and 200 μL MgSO_4_ [1 mol∙L^−1^] (E. Merck; Darmstadt, Germany) were added into a flask and filled with water to 100 mL, serving as the minimum medium (MM).

For the fungal test, agar from Agar Technical BD Difco was used in a 14 wt% water solution as the substrate for the Petri dishes. The inoculated fungi were *Trametes versicolor* and *Agrocybe aegerita* collected in local forests and stored in a protected dark environment in the laboratory of the TESAF department of the University of Padua. The reactivation of the mycelia was conducted by pouring them into a water–agar substrate for 4 weeks.

### 2.2. Methods

#### 2.2.1. Bacteria Tests

Blocks of tannin–furanic foams (TFF) were exposed to a compost heap to find bacteria colonies more likely to tolerate or even degrade the TFP. For bacterial growth assays bacterial strains were isolated from blocks of 15 × 15 × 15 mm^3^ standard tannin–furanic foam (TFF), which were kept in a litter and in a compost heap of the Botanical Garden of the University of Marburg for 21 days. After exposure, foam blocks were transferred into 20 mL of phosphate-buffered saline and sonicated for 7 min in order to detach bacteria from the foam. Then, 1, 10, 20 and 40 µL of the resulting suspension was transferred to agar plates containing 20 mL of the standard medium (SM). After 3 days, individual colonies were picked and isolated on a separate agar plate. Two strains from the families *Bacillaceae* and *Pseudomanadaceae* were identified: The 16S rRNA gene (target regions V3, V4, ~630 bp) was amplified and sequenced (forward and reverse) with two universal primers ALer1_341f (5′-CCT ACG GGA GGC AGC AG-3′) and Buniv_970r (5′-CCG TCA ATT CMT TTG AGT TT -3′) with standard PCR-conditions (95 °C for 2 min: 1 cycle; 94 °C for 30 s, 53 °C for 30 s, 72 °C for 40 s: 35 cycles). After purification with the MSB Spin PCRapace kit (Stratec molecular; Berlin, Germany), the PCR-products were sent to Eurofins Genomics (Ebersberg, Germany) for sequencing. Raw sequences of the 16S rRNA genes are available at the NCBI Sequence Read Archive (SRA) under the BioProject accessions SAMN26875395 (*Bacillaceae*) and SAMN26875394 (*Pseudomonadaceae*).

Bacterial growth was investigated in the minimum media (MM) with the addition of 1% MT or TFP containing either non or 0.1% glucose. To these solutions and suspensions bacteria were added at an initial optical density (OD) of OD = 1 and incubated at 36 °C in 96-well plates (Thermo Fisher Scientific, Dreieich, Germany). Bacterial growth was tracked for 6 days in a plate reader (CLARIOstar Plus, BMG LABTECH GmbH, Ortenberg, Germany) that measured the OD in each well at an interval of 30 min.

#### 2.2.2. Fungal Tests

A 14% water–agar solution was prepared and sterilized at 120 ± 1 °C for 15 min. While still warm (~40 °C), the water–agar solution was homogeneously distributed applying 10 mL to each of the 144 Petri dishes (d = 75 mm). Then, 30 mL of solutions of mimosa tannin (MT) and suspensions of tannin–furanic polymer (TFP) with water were prepared at different concentrations (wt%): 0.1%; 1%; 10%; 20%.

0.5 mL of solution (MT) or suspension (TFP) were applied on each Petri dish containing hardened water–agar and quickly distributed on the surface with a glass spreader. Eight Petri dishes were plated for each formulation. The added solution was dried at room temperature for one day before inoculation. Control Petri dishes were prepared without adding anything else on the 10 mL of water–agar solution.

The inoculation was performed by laying a mycelium plug of the fungus 7 mm in diameter in the center of the Petri dishes. Eight repetitions for each fungal strain were conducted for every concentration. Summarizing the design of experiments: two substrates (MT and TFP), four concentrations (and one control), two fungi and eight repetitions for a total of one-hundred and forty-four tests.

The whole set of experiments was maintained at room temperature and constantly monitored. Periodic checks of the fungal development were performed by tracking the area of fungal growth after 4, 7, 10, 12, 14, 17, 20, 26 and 32 days. The diameters per each colony were measured along three axes, subtracting the diameter of the original mycelium plug.

### 2.3. Statistical Analysis

Statistical analyses on bacterial growth was performed using R Statistical Software, version 3.6.0 [28]. The maximum possible population size in a particular environment, referred to as carrying capacity k, was determined for each growth curve using the R CRAN Packages “growthCurver” [29]. Carrying capacity k of both strains was compared across the media applying an ANOVA followed by a Tukey post-hoc test.

The analyses of the data collected on white-rot fungi growth was performed using R Statistical Software, version 3.6.0. The R/CRAN Packages used were “tidyverse”, “ggpubr” and “rstatix” [28,30,31,32]. Data of daily fungal growth rate over time was computed for each fungus by subtracting the previous from the following diametrical measurements and expressed as cm∙day^−1^. All data firstly underwent the Shapiro–Wilk’s Normality Test. The daily growth rates of the fungi, per each additive and concentrations, were then elaborated by non-parametric Kruskal–Wallis (*p* < 0.05) to evaluate the presence of significant differences among groups, followed by multiple comparisons between groups using Dunn’s Test (*p* < 0.05).

## 3. Results

The exposure of bacteria and white-rot fungi to mimosa tannin (MT) powder and to grinded tannin–furanic polymer (TFP) allowed to monitor the effect of these substances on the growth of these microorganisms. 

### 3.1. Bacteria Exposure Tests

After tracking the optical density for the different formulations, the carrying capacity k was calculated for comparing microbial growth characteristics.

In Figure 3, the growth of *Bacillaceae* in the presence of tannin and tannin–furanic powder, with and without extra glucose, is presented as the carrying capacity.

Without glucose, the strain showed a significant increase in growth for the tannin–furanic polymer, comparable to that of glucose alone (more than two-fold). MT and TFP in combination with glucose were not growth inhibitory and even increased significantly for TFP. This bacterium seems to be able to take advantage of TFP with an additional food source and even solely from its polymerized macromolecule, whereas MT did not show any influence on bacterial growth.

In Figure 4, the growth of *Pseudomonaceae* is reported. Here, different trends are observable. Compared to the control, the presence of MT and TFP does not enhance the growth of bacteria when no glucose is added. Conversely, when glucose is added, the presence of MT and TFP significantly enhances the growth of the *Pseudomonaceae* strain compared to glucose alone.

The bacterial test showed, that in both species, tannin without glucose resulted in no growth enhancement, whereas the addition of glucose could trigger a synergistic effect, and the bacteria grew better than with glucose alone. Similar observations were made for TFP (with the exception of *Bacillaceae*), which seems to be able to exploit the polymer as a carbon source as it does with glucose.

### 3.2. White-Rot Fungal Exposure Tests

The colony diameters observed over time for the two fungi *T. versicolor* and *A. aegerita* have shown comparable trends, which are shown in Figure 5.

Despite the two fungi showing different growth rates, low concentrations (0.1%) of tannin and tannin–furanic polymers tend to allow a faster growth of the mycelium, while at high concentrations (10%), the growth is lower or even completely inhibited. In particular, it can be noticed that the inhibitory effect against *T. versicolor* at 10 and 20% is higher for tannin alone. Conversely, *A. aegerita* was completely inhibited for the full period of 32 days at 10% concentration by both MT and TFP but reached a bigger colony diameter after the whole period for the TFP at 1%.

The observed effect of MT and TFP after 7 days for the strain *T. versicolor,* according to its daily growth rate, is reported in Figure 6.

In these graphics, the growth effect for small concentrations of MT (Figure 6a) and TFP (Figure 6b) is highlighted along with the inhibitory effect at higher concentrations. The observed trends for TFP and MT after 7 days from the inoculation are very similar; however, the observed growth at 10% TFP suggests that once polymerized, tannin loses some of its inhibitory character on *Trametes versicolor*. Low doses up to 1% have a growth-enhancing effect and show no inhibition.

The effect on *A. aegerita* which grows slowly and could therefore be comparatively monitored until day 17, was slightly different (Figure 7).

Additionally, for *A. aegerita,* small concentrations of 0.1% TFP and MT tend to speed up mycelia growth, though not significantly. Unlike *Trametes versicolor*, when already at 1%, slight growth inhibition becomes significant for TFP, but is comparable to the behavior of MT. High concentrations of 10 and 20% for MT and TFP lead to complete growth inhibition. Overall, this strain of fungi appears more sensitive to MT and TFP than *T. versicolor*. It can be stated that both MT and TFP inhibit mycelial growth at high concentrations, while low concentrations tend to favor mycelial growth, and that TFP seems to be less inhibitory than MT towards *Trametes versicolor*.

## 4. Discussion

Due to their antimicrobial properties, tannins are considered recalcitrant to degradation, but their digestion occurs frequently in nature. We can observe two contrasting effects in the microorganism growth: (i) feeding and (ii) hindering effect.

The feeding effect occurs when the microorganism can exploit the molecule to recover energy, for instance, hydrolysable tannin can be reinserted in the citric acid cycle after relatively simple breakdown (aerobic/anaerobic) by tannase enzymes, while condensed tannins, such as the mimosa extract, require different reductive breakdowns, which render them even more recalcitrant [24,33]. However, certain microorganisms such as white-rot fungi, including *Trametes versicolor* and bacteria of the family *Pseudomonaceae* and *Bacillaceae,* were reported to degrade condensed tannins or catechin, the basic constituent of condensed tannins [24]. For the degradation of condensed tannins instead of tannase, both fungi and bacteria use their phenoloxidase system involving enzymes such as laccase, peroxidase or catechin oxygenase [34,35,36,37]. In vitro experiments of other catechin degrading fungi also show that this effect seems to only take place when exposed to low concentrations [34].

The hindering effect occurs when the presence of tannin inhibits the role of other enzymes of the microorganism. There are several studies that report this observation for many enzymes such as α-amylase, α-glucosidase, glucoamylase, cholinesterase and tyrosinase [38,39,40,41]. The mechanism occurring is the formation of stable tannin–protein complexes, which hinder the execution of their enzymatic functions [42,43].

The carrying capacity of media was increased by the TFP without glucose for *Bacillaceae,* indicating the ability to use the tannin–furanic polymer as a carbon source. Mimosa tannin does not significantly promote the growth of the bacteria without glucose, but as soon as glucose is added it can contribute to further enhance the carrying capacity supposing the occurrence of a synergic effect in the metabolisms, which is generally known to possibly occur in bacterial growth exposed to multiple carbon sources [44,45].

*T. versicolor* and *A. aegerita* grew well when exposed to a limited concentration of MT, confirming the finding of the literature that certain limited doses even enhance fungal growth in certain species of white- and soft-rot fungi [46]. When the concentration of tannin increases, specific enzymes of the fungi are inhibited and the mycelium does not grow [47,48,49].

Within the low doses, no significant difference could be found for MT and TFP, but for 10 and 20% TFP, the higher tolerance of *T. versicolor* must be emphasized, suggesting that the polymerized tannin has a lower inhibitory effect against this strain than the mimosa tannin. Conversely, *A. aegerita* does not grow for high concentration of TFP, which means that, for this fungus, the inhibitory effect of tannin also remains relatively unaltered after polymerization.

The good tolerance or even growth-enhancing properties at low concentrations of MT and TFP for the fungi and bacteria tested increases further interest in the suggested use as an ammonia mitigator in agriculture since studies showed their effectiveness already in very low amounts, where 1 w% of the polymer in 250 mL of 1:1 diluted cattle slurry led to a reduction in gaseous emissions by 60% [21].

The results of this study can be summarized as follows. In comparison to MT, the TFP registers (i) a similar feeding and slightly milder inhibition effect towards tested fungi, (ii) higher bacterial growth in *Bacillaceae* and (iii) similar behavior in *Pseudomonaceae* and *Agrocybe aegerita.*

At low concentrations, fungi can tolerate tannins (MT) and TFP or even benefit from their presence to grow faster, whereas bacteria were already inhibited without additional glucose, except *Bacillaceae,* which showed enhanced growth towards TFP. Findings in various literature suggest that such types of fungi and bacteria may even metabolize condensed tannins and, thus, TFP, by breaking them down through different oxygenases instead of tannase enzymes, before being reinserted into the citric acid cycle (feeding effect) [24]. However, this design cannot answer this question satisfactorily as it is known that tannin and its furanic polymer also contain small amounts of sugars, which could also be responsible for the enhanced growth [7,46]. In contrast, this study showed that TFP had slightly lower antimicrobial effects on the growth of tannin-resistant fungi compared to the natural extract. When the concentration was increased, the inhibitory effect of the polyphenols against other enzymes becomes dominant and the fungi did not grow. Considering that the tested bacteria and fungi are known to have good tolerance to tannin extracts, the antimicrobial properties found could also be interesting for the development of antimicrobial bio-plastics with potential use cases such as hygienic surfaces in hospitals. Further studies for the standardized evaluation of its antibacterial activity with specific bacteria (*Staphylococcus aureus, Escherichia coli*.) according to ISO 22196:2011 would, therefore, be particularly interesting. Although the aim of the study was not to evaluate biodegradability, it is worth highlighting the growth of *Bacillaceae* with TFP, which recorded comparable growth values as the glucose reference sample.

## 5. Conclusions

This research work evaluated the influence of a tannin–furanic polymer compared to mimosa tannin on the in vitro growth of two strains of bacteria of the family *Bacillaceae* and *Pseudomonaceae and* two white-rot fungi (*Trametes versicolor* and *Agrocybe aegerita)*. Tested bacteria and fungi were selected for the ability to tolerate or eventually degrade condensed tannins. The impact of the tannin content was evaluated for the fungal experiments while the selected concentration (1%) for bacteria gave complementary information to this study. In bacteria, the mimosa tannin, known for its antimicrobial properties, led to higher bacteria growth when it was combined with glucose compared to with glucose alone but did not show any growth enhancement without additional glucose. Aside from the synergistic effect occurring on the bacterial growth through the combination of glucose with MT/TFP, the bacteria of the family *Bacillaceae* even seems to be able to solely use TFP as a carbon source, as it shows a similar increase in growth as when glucose alone serves as the food source. In the white-rot fungi, MT and TFP enhanced mycelia growth for low concentrations up to 1%, whereas doses of 10% and higher clearly inhibited fungi growth. The influence of TFP on fungi and bacteria was very similar to that of MT, however showing slightly less inhibitory effect on the tested fungi *Trametes versicolor* and bacteria *Bacillaceae*. These results suggest that the tannin–furanic polymer loses some of its inhibitory effect on microorganisms compared to the natural mimosa tannin extract, maintaining similar antimicrobial effects on microorganisms as tannin. These results might be particularly interesting in the design of antimicrobial tannin-based bio-plastics or for supporting the use of these materials as ammonia mitigator in agriculture.

## Figures and Tables

**Figure 1 polymers-15-00175-f001:**
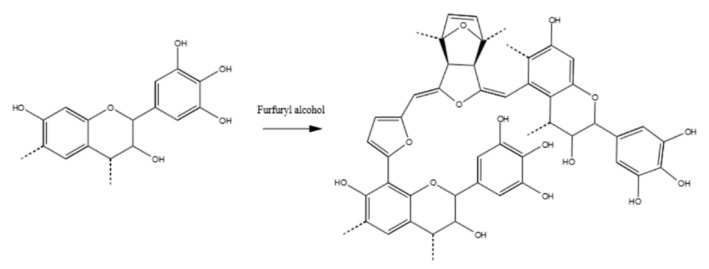
Possible reaction product of tannin–furfuryl alcohol polymers [8].

**Figure 2 polymers-15-00175-f002:**
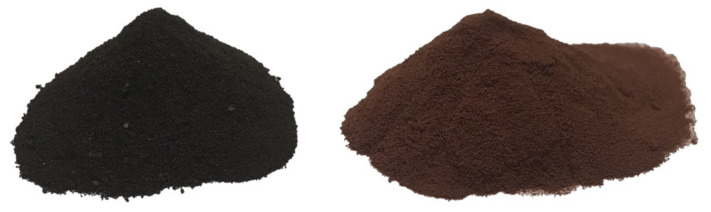
Powder of tannin–furanic polymer (TFP) (**left**) and mimosa tannin extract (MT) (**right**).

**Figure 3 polymers-15-00175-f003:**
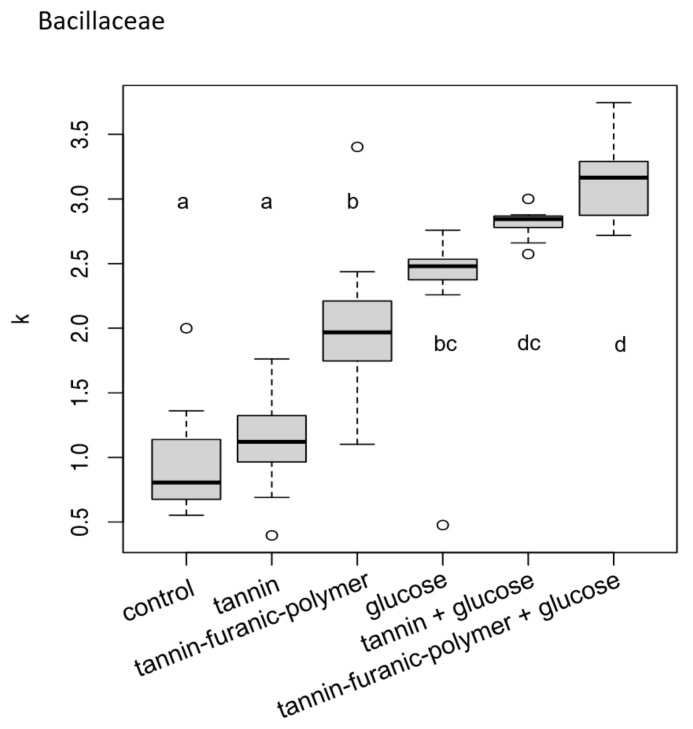
Growth of the *Bacillaceae* in presence of tannin and tannin–furanic polymer with and without extra glucose. Carrying capacity of the *Bacillaceae* family differed between media (ANOVA: F5,66 = 53.62, *p* < 0.001). Different letters on top or below boxplots indicate significant differences in carrying capacity k between media.

**Figure 4 polymers-15-00175-f004:**
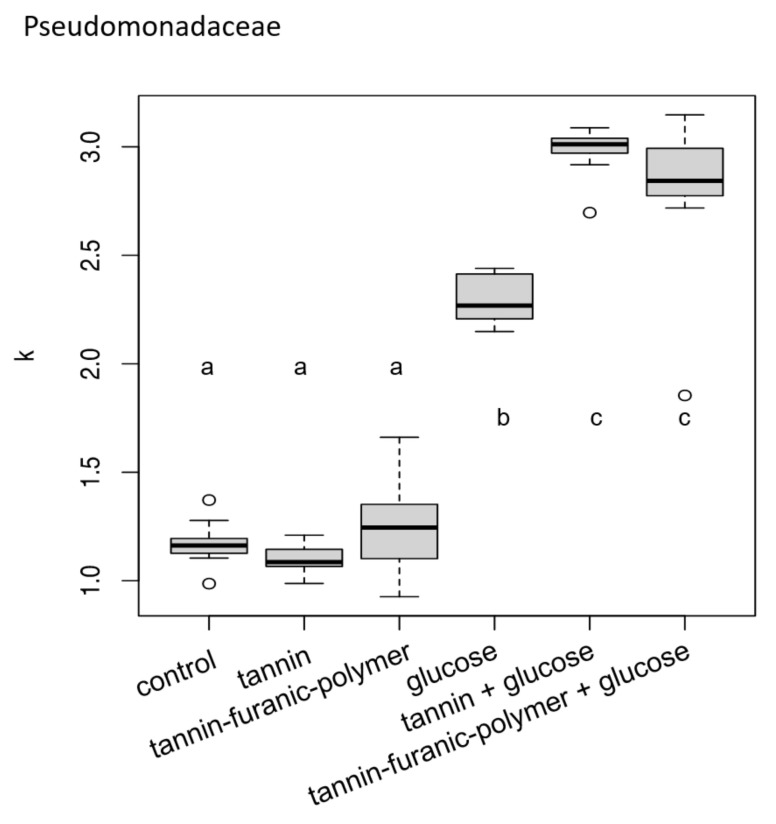
Growth of the *Pseudomonaceae* in presence of tannin and tannin–furanic polymer with and without extra glucose. Carrying capacity of the Pseudomonadaceae strain differed between media (ANOVA: F5,66 = 290.6, *p* < 0.001). Different letters on top or below boxplots indicate significant differences in carrying capacity k between media.

**Figure 5 polymers-15-00175-f005:**
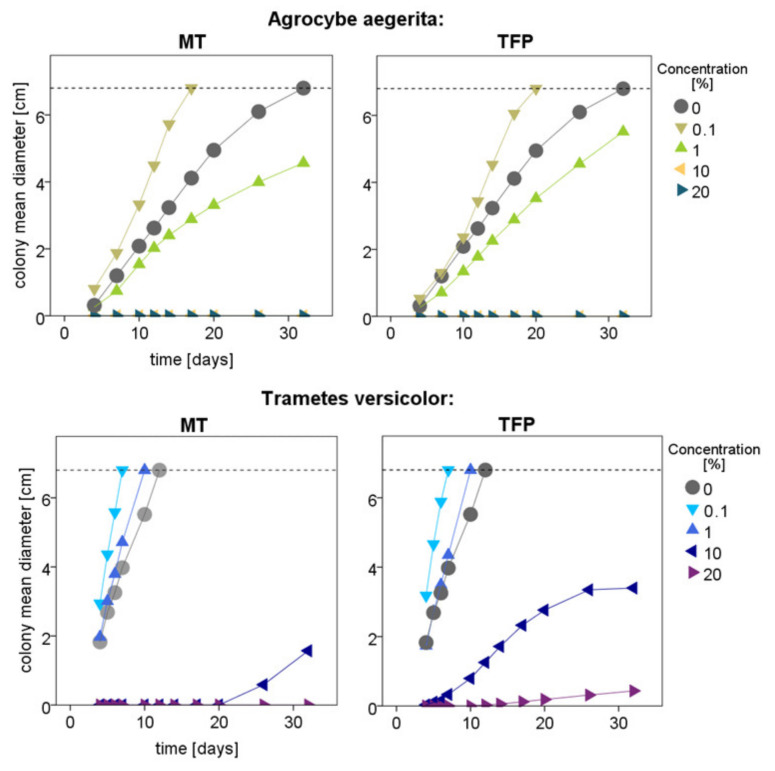
Mean mycelium diameter without the initial plug in cm of the two fungi during the 32 days of exposure.

**Figure 6 polymers-15-00175-f006:**
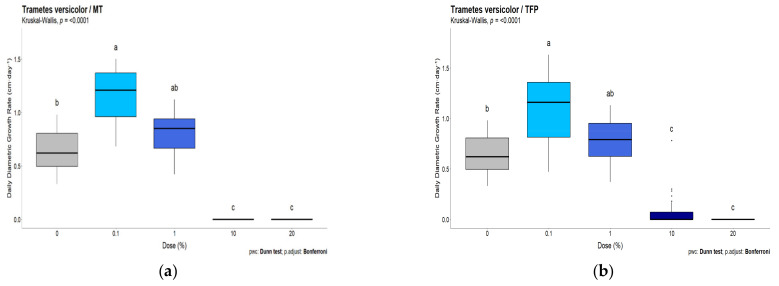
Daily growth rate up to day 7 for *T. versicolor* at different concentrations of (**a**) mimosa tannin and (**b**) tannin–furanic polymer.

**Figure 7 polymers-15-00175-f007:**
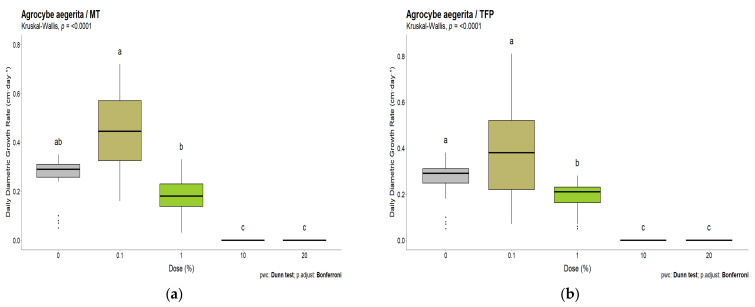
Daily growth rate for *A. aegerita* at different concentrations of (**a**) mimosa tannin (17 days) and (**b**) tannin–furanic polymer (20 days).

## Data Availability

The data acquired during this study are available by the corresponding author. The original elaborations for bacteria are available by R. R. Junker while the ones on fungal tests are stored by G. Tondi.

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
