# Peer review of "Effect of Tannin Furanic Polymer in Comparison to Its Mimosa Tannin Extract on the Growth of Bacteria and White-Rot Fungi"

_polymers, 2022, doi:10.3390/polym15010175_

Round 1

Reviewer 1 Report

Research on the antibacterial and fungicidal activity of tannin-furanic polymer was evaluated. The Authors compared the biocidal effect of tannin-furanic polymer to the biocidal effect of tannin extract.

The methodological actions taken require some additions and explanations.

The selection of microorganisms is a bit unusual. On the one hand, research is conducted on organisms (bacteria) isolated from the soil, which is a good direction in research determining the ability of microorganisms to biodegrade (bioutilization). On the other hand, the Authors use fungi that cause white decay of wood, indicating that these are fungi resistant to tannins and even capable of biodegrading them, but what is the purpose of using such microorganisms. Why were fungi isolated from the soil not used? Why was the effect of tannins on soil bacteria and fungi causing white decay of wood investigated?

The work should define the future applications of such a polymer and direct research, including the selection of microorganisms, to such an application. If the polymer was to be used for the production of bioplastic with antimicrobial properties, it is worth choosing microorganisms proposed in standards, e.g. ISO 22196 or EN 14885.

If the polymer, as suggested by the Authors of the paper, would be used in wood protection, it would be appropriate to conduct tests against at least one white rot fungus (e.g. T. versicolor) and at least one brown rot fungus (e.g. C. puteana) and mold fungi (e.g. . Ch. globosum). On the other hand, in the protection of wood, bacteria are not so important, unless the protection concerns archaeological wood.

Therefore, changes should be made to the methodology or appropriate changes should be made to the purpose and scope of the research.

Line 126 - "T. versicolor" should be written in italics

Line 120 NH4Cl "4" should be subscript

Line 158 - "30 ml of solutions of mimosa tannin" - what solvent was used to dissolve mimosa tannin

Line 159 - "(MT) and suspensions of tannin-furanic polymer" in which a fragmented fraction of tannin-furanic polymer was suspended

Line 237 - correct "figure 4" to "Figure 4"

Line 81 correct "Polyfurfuryl" to "polyfurfuryl"

It should be indicated in Figure 4 which graph is 4a and which is 4b, similarly, Figure 5 should be corrected.

Figure 1 - please check the statistics again because from the graph it seems that the effect of furanic polymer tannins on bacterial growth compared to tannins in the statistical test should give an "ab" result.

Author Response

Please, refer to the attached document

Reviewer 2 Report

The Authors showed the examinations of the impact of tannins on selected bacteria strains and fungi. Unfortunately, in the presented work there are no detailed studies of how the samples were made, as well as the possible impact of the tannin content on the presented results.

Furthermore, the Author even did not prove that obtained tannins. Additional tests are necessary. What is more, the effect of the tannin concentration is not presented here. Even detailed sample composition was not presented.

It should be highlighted, that the polymerization process was not described - an additional figure, where the final structural components will be presented is necessary.

Unfortunately, my recommendation is rejected.

Detailed comments are listed below:

For that reason line 12 - this statement is not true. Some papers show the compositions with TA. Furthermore, what do you mean here by polymerization - TA in polymer matrix?

Furthermore, the hypothesis is the opposite of other studies which showed that tannins exhibit antimicrobial properties. Here, additional studies of the tannin composition are needed i.e. to evaluate the concentration of the main compounds.

Moreover, the sample preparation description is not meaningful. Starting from, the abstract, through the introduction and materials and methods it is difficult to catch what was truly studied.

Please clearly state did you extracted tannins from the mimosa.

Probably the effect of low antimicrobial activity was due to copolymerization

Please add some comments on why you polymerized tannins.

The description of the methods is also confusing. First, you stated that the powders were obtained, now in section 2, it is mentioned that foams. Please clarify it.

Why no verification of the structure and composition tests were done?

Results and discussion

- in this section, the effect of the tannin concentration was not presented

- how are you sure that the amount of tannin was the same in the presented sample?

- additional structural tests are mandatory.

Author Response

Please, refer to the attached document.

Reviewer 3 Report

Dear Editor,

Thank you for giving me the opportunity to revise the manuscript entitled “Effect of tannin furanic polymer in comparison to its mimosa tannin extract on the growth of bacteria and white rot fungi” by Jonas Eckardt and his/her colleagues that was submitted to “polymers”. The MS submitted is suitable for polymers, and some interesting results were showed. The manuscript provided appropriate information about the studied task, but there are several requirements that have to consider by the authors. 

Comment 1: Please check the format of the full text carefully. Such as Line 102, Line 129 the excess '. ' should be deleted.

Comment 2: Line 203 The sentence “Without glucose, the strain showed a significant increase in growth for the tannin furanic polymer, comparable to that of glucose alone (more than two-fold).” should be described in greater detail and accuracy.

Comment 3: The discussion section should be carefully revised.

Comment 4: Please check the format of the references carefully. Such as Line 388, Line 434, 

Best regards,

Author Response

Please, refer to the attached document.

Round 2

Reviewer 1 Report

The work on the influence of tannins and tannin furanic polymer on the growth of selected fungi and bacteria was submitted for re-evaluation.

After making some additions by the Authors of the work, some of the provisions in the article should be revised again so that it can be published.

The Authors of the paper describe that their goal was not to assess biodegradation by fungi, but descriptions such as: " Also the synergic effect in bacterial growth occurring for MT and TFP when glucose is added, indicates that species involved in the degradation of tannins might also degrade tannin furanic polymers especially in a more complex media like natural soil or specific composting setups" mislead the reader. In this case, either reference should be made to the relevant literature or the statement should be deleted, as the Authors of the studies did not carry out experiments showing that tannin-based polymers are biodegradable by the indicated microorganisms.

A similar description in the conclusions (line: 375-376) needs to be edited, because the presented test results do not allow to find out whether the tannins and tannins of the furanic polymer are biodegradable.

Please indicate how the control sample was prepared. The control indicated in Fig. 6b should be double. First, control of fungal growth on medium alone and control of fungal growth on tannin-free polymer. This type of control will allow to demonstrate the effect of pure furanic polymer on the growth of fungi.

The results of the work show the effect of tannins on the growth of bacteria and fungi, but it does not correspond to the content contained in the Introduction. The Authors of the research largely focus on the description of biodegradation (lines 64-82), which was not the subject of research. Such a long introduction misleads the reader as it implies that biodegradation studies will be the subject of the article.

The introduction of the work should be redrafted, as I wrote in the previous review. It should refer to the conducted research. Biodegradation can be mentioned, but this topic is not related to the purpose or scope of the work.

Author Response

Dear editor, dear reviewer, 

Please find in attachment the detailed answer to your comments,

Thanks and best regards.

Reviewer 2 Report

The Authors strongly improved the manuscript content.

Still, minor revision is needed.

A lot of important information was presented in the response to Reviewers. Please include in the main text:

- the information: "The impact of the tannin content was evaluated for the fungal experiments while the selected concentration for bacteria gave complementary information" - include in the abstract and conclusions as well (please keep in mind what readers usually read)

- The tannin extract that was used in this study was of industrial origin, and commercially available (mimosa from the company Tanac). The detailed composition of the tannin extract could be found in the safety data sheet of the extract, however detailed spectroscopic and chromatography analysis were done in former years in the work of our group, however, we did not cite them because related to other objectives. - this comment inserted in materials and methods

- the statement that you investigated powders should be highlighted. Furthermore, an additional image of the samples will strongly increase the manuscript content.

Just to be sure, did you focus in this paper only on antimicrobial activity or some other aspects such as obtained powder characterization (structure i.e. SEM, DLS, etc.)?

Author Response

(The authors gave the same response as above.)

Round 3

Reviewer 1 Report

The work has been largely improved.

However, when reading the "Introduction" chapter, there is still no reference to the purpose of the work and the title of the work. The Authors of the study removed the part not related to the subject, but the content of this chapter lacks reference to the results of other researchers who dealt with similar topics.

Since the Authors of the paper evaluated the effect of tannins on the growth of fungi and bacterial activity, they should focus at least a little on the description of similar results obtained by other authors and emphasize the novelty of their work.

Author Response

Dear reviewer 1,

We appreciate that the previously submitted version was improved and we thank you again for the suggestions.

 In this latest version we have added in the introduction a paragraph were the interaction of microorganisms with condensed tannin extract is reported. 

Yours sincerely,

Gianluca Tondi, 

On behalf of the co-authors team
